# FairFil: Contrastive Neural Debiasing Method for Pretrained Text Encoders

**Pengyu Cheng**[∗] , **Weituo Hao**[∗], **Siyang Yuan** , **Shijing Si** , **Lawrence Carin**
Department of Electrical and Computer Engineering, Duke University
{pengyu.cheng,weituo.hao,siyang.yuan,shijing.si,lcarin}@duke.edu

## Abstract

Pretrained text encoders, such as BERT, have been applied increasingly in various natural language processing (NLP) tasks, and have recently demonstrated significant performance gains. However, recent studies have demonstrated the existence of *social bias* in these pretrained NLP models. Although prior works have made progress on word-level debiasing, improved sentence-level fairness of pretrained encoders still lacks exploration. In this paper, we proposed the *first* neural debiasing method for a pretrained sentence encoder, which transforms the pretrained encoder outputs into debiased representations via a fair filter (FairFil) network. To learn the FairFil, we introduce a contrastive learning framework that not only minimizes the correlation between filtered embeddings and bias words but also preserves rich semantic information of the original sentences. On real-world datasets, our FairFil effectively reduces the bias degree of pretrained text encoders, while continuously showing desirable performance on downstream tasks. Moreover, our *post hoc* method does not require any retraining of the text encoders, further enlarging FairFil's application space.

## 1 Introduction

Text encoders, which map raw-text data into low-dimensional embeddings, have become one of the fundamental tools for extensive tasks in natural language processing (Kiros et al., 2015; Lin et al., 2017; Shen et al., 2019; Cheng et al., 2020b). With the development of deep learning, large-scale neural sentence encoders pretrained on massive text corpora, such as Infersent (Conneau et al., 2017), ELMo (Peters et al., 2018), BERT (Devlin et al., 2019), and GPT (Radford et al., 2018), have become the mainstream to extract the sentence-level text representations, and have shown desirable performance on many NLP downstream tasks (MacAvaney et al., 2019; Sun et al., 2019; Zhang et al., 2019). Although these pretrained models have been studied comprehensively from many perspectives, such as performance (Joshi et al., 2020), efficiency (Sanh et al., 2019), and robustness (Liu et al., 2019), the *fairness* of pretrained text encoders has not received significant research attention.

The fairness issue is also broadly recognized as *social bias*, which denotes the unbalanced model behaviors with respect to some socially sensitive topics, such as gender, race, and religion (Liang et al., 2020). For data-driven NLP models, social bias is an intrinsic problem mainly caused by the unbalanced data of text corpora (Bolukbasi et al., 2016). To quantitatively measure the bias degree of models, prior work proposed several statistical tests (Caliskan et al., 2017; Chaloner & Maldonado, 2019; Brunet et al., 2019), mostly focusing on word-level embedding models. To evaluate the sentence-level bias in the embedding space, May et al. (2019) extended the Word Embedding Association Test (WEAT) (Caliskan et al., 2017) into a Sentence Encoder Association Test (SEAT). Based on the SEAT test, May et al. (2019) claimed the existence of social bias in the pretrained sentence encoders.

Although related works have discussed the measurement of social bias in sentence embeddings, debiasing pretrained sentence encoders remains a challenge. Previous word embedding debiasing methods (Bolukbasi et al., 2016; Kaneko & Bollegala, 2019; Manzini et al., 2019) have limited assistance to sentence-level debiasing, because even if the social bias is eliminated at the word level,

---

[∗]Equal contribution.

the sentence-level bias can still be caused by the unbalanced combination of words in the training text. Besides, retraining a state-of-the-art sentence encoder for debiasing requires a massive amount of computational resources, especially for large-scale deep models like BERT (Devlin et al., 2019) and GPT (Radford et al., 2018). To the best of our knowledge, Liang et al. (2020) proposed the only sentence-level debiasing method (Sent-Debias) for pretrained text encoders, in which the embeddings are revised by subtracting the latent biased direction vectors learned by Principal Component Analysis (PCA) (Wold et al., 1987). However, Sent-Debias makes a strong assumption on the linearity of the bias in the sentence embedding space. Further, the calculation of bias directions depends highly on the embeddings extracted from the training data and the number of principal components, preventing the method from adequate generalization.

In this paper, we proposed the first neural debiasing method for pretrained sentence encoders. For a given pretrained encoder, our method learns a fair filter (FairFil) network, whose inputs are the original embeddings of the encoder, and outputs are the debiased embeddings. Inspired by the multi-view contrastive learning (Chen et al., 2020), for each training sentence, we first generate an augmentation that has the same semantic meaning but in a different potential bias direction. We contrastively train our FairFil by maximizing the mutual information between the debiased embeddings of the original sentences and corresponding augmentations. To further eliminate bias from sensitive words in sentences, we introduce a debiasing regularizer, which minimizes the mutual information between debiased embeddings and the sensitive words' embeddings. In the experiments, our FairFil outperforms Sent-Debias (Liang et al., 2020) in terms of the fairness and the representativeness of debiased embeddings, indicating our FairFil not only effectively reduces the social bias in the sentence embeddings, but also successfully preserves the rich semantic meaning of input text.

## 2 PRELIMINARIES

Mutual Information (MI) is a measure of the "amount of information" between two variables (Kullback, 1997). The mathematical definition of MI is

$$\mathcal{I}(\boldsymbol{x}; \boldsymbol{y}) := \mathbb{E}_{p(\boldsymbol{x},\boldsymbol{y})}\Big[ \log \frac{p(\boldsymbol{x},\boldsymbol{y})}{p(\boldsymbol{x})p(\boldsymbol{y})} \Big], \tag{1}$$

where $p(\boldsymbol{x}, \boldsymbol{y})$ is the joint distribution of two variables $(\boldsymbol{x}, \boldsymbol{y})$, and $p(\boldsymbol{x}), p(\boldsymbol{y})$ are respectively the marginal distributions of $\boldsymbol{x}, \boldsymbol{y}$. Recently, mutual information has achieved considerable success when applied as a learning criterion in diverse deep learning tasks, such as conditional generation (Chen et al., 2016), domain adaptation (Gholami et al., 2020), representation learning (Chen et al., 2020), and fairness (Song et al., 2019). However, the calculation of exact MI in (1) is well-recognized as a challenge, because the expectation $w.r.t$ $p(\boldsymbol{x}, \boldsymbol{y})$ is always intractable, especially when only samples from $p(\boldsymbol{x}, \boldsymbol{y})$ are provided. To this end, several upper and lower bounds have been introduced to estimate the MI with samples. For MI maximization tasks (Hjelm et al., 2018; Chen et al., 2020), Oord et al. (2018) derived a powerful MI estimator, InfoNCE, based on noise contrastive estimation (NCE) (Gutmann & Hyvärinen, 2010). Given a batch of sample pairs $\{(\boldsymbol{x}_i, \boldsymbol{y}_i)\}_{i=1}^N$, the InfoNCE estimator is defined with a learnable score function $f(\boldsymbol{x}, \boldsymbol{y})$:

$$\mathcal{I}_{\text{NCE}} := \frac{1}{N}\sum_{i=1}^{N} \log \frac{\exp(f(\boldsymbol{x}_i, \boldsymbol{y}_i))}{\frac{1}{N}\sum_{j=1}^{N}\exp(f(\boldsymbol{x}_i, \boldsymbol{y}_j))}. \tag{2}$$

For MI minimization tasks (Alemi et al., 2017; Song et al., 2019), Cheng et al. (2020a) introduced a contrastive log-ratio upper bound (CLUB) based on a variational approximation $q_\theta(\boldsymbol{y}|\boldsymbol{x})$ of conditional distribution $p(\boldsymbol{y}|\boldsymbol{x})$:

$$\mathcal{I}_{\text{CLUB}} := \frac{1}{N}\sum_{i=1}^{N}\Big[ \log q_\theta(\boldsymbol{y}_i|\boldsymbol{x}_i) - \frac{1}{N}\sum_{j=1}^{N} \log q_\theta(\boldsymbol{y}_j|\boldsymbol{x}_i) \Big]. \tag{3}$$

In the following, we use the above two MI estimators to induce the sentence encoder, eliminating the biased information and preserving the semantic information from the original raw text.

## 3 METHOD

Suppose $E(\cdot)$ is a pretrained sentence encoder, which can encode a sentence $\boldsymbol{x}$ into low-dimensional embedding $\boldsymbol{z} = E(\boldsymbol{x})$. Each sentence $\boldsymbol{x} = (w^1, w^2, \dots, w^L)$ is a sequence of words. The embedding space of $\boldsymbol{z}$ has been recognized to have social bias in a series of studies (May et al., 2019; Kurita

Table 1: Examples of generating an augmentation sentence under the sensitive topic *"gender"*.

|  | Bias direction | Sensitive Attribute words | Text content |
|---|---|---|---|
| Original | male | he, his | {He} is good at playing {his} basketball. |
| Augmentation | female | she, her | {She} is good at playing {her} basketball. |

et al., 2019; Liang et al., 2020). To eliminate the social bias in the embedding space, we aim to learn a fair filter network $f(\cdot)$ on top of the sentence encoder $E(\cdot)$, such that the output embedding of our fair filter $\boldsymbol{d} = f(\boldsymbol{z})$ can be debiased. To train the fair filter, we design a multi-view contrastive learning framework, which consists of three steps. First, for each input sentence $\boldsymbol{x}$, we generate an augmented sentence $\boldsymbol{x}'$ that has the same semantic meaning as $\boldsymbol{x}$ but in a different potential bias direction. Then, we maximize the mutual information between the original embedding $\boldsymbol{z} = f(\boldsymbol{x})$ and the augmented embedding $\boldsymbol{z}' = f(\boldsymbol{x}')$ with the InfoNCE (Oord et al., 2018) contrastive loss. Further, we design a debiasing regularizer to minimize the mutual information between $\boldsymbol{d}$ and sensitive attribute words in $\boldsymbol{x}$. In the following, we discuss these three steps in detail.

### 3.1 DATA AUGMENTATIONS WITH SENSITIVE ATTRIBUTES

We first describe the sentence data augmentation process for our FairFil contrastive learning. Denote a social sensitive topic as $\mathcal{T} = \{\mathcal{D}_1, \mathcal{D}_2, \ldots, \mathcal{D}_K\}$, where $\mathcal{D}_k$ ($k = 1, \ldots, K$) is one of the potential bias directions under the topic. For example, if $\mathcal{T}$ represents the sensitive topic *"gender"*, then $\mathcal{T}$ consists two potential bias directions $\{\mathcal{D}_1, \mathcal{D}_2\} = \{$*"male"*, *"female"*$\}$. Similarly, if $\mathcal{T}$ is set as the major *"religions"* of the world, then $\mathcal{T}$ could contain $\{\mathcal{D}_1, \mathcal{D}_2, \mathcal{D}_3, \mathcal{D}_4\} = \{$*"Christianity"*, *"Islam"*, *"Judaism"*, *"Buddhism"*$\}$ as four components.

For a given social sensitive topic $\mathcal{T} = \{\mathcal{D}_1, \ldots \mathcal{D}_K\}$, if a word $w$ is related to one of the potential bias direction $\mathcal{D}_k$ (denote as $w \in \mathcal{D}_k$), we call $w$ a *sensitive attribute word* of $\mathcal{D}_k$ (also called bias attribute word in Liang et al. (2020)). For a sensitive attribute word $w \in \mathcal{D}_k$, suppose we can always find another sensitive attribute word $u \in \mathcal{D}_j$, such that $w$ and $u$ has the equivalent semantic meaning but in a different bias direction. Then we call $u$ as a *replaceable word* of $w$ in direction $\mathcal{D}_j$, and denote as $u = r_j(w)$. For the topic *"gender"* = $\{$*"male"*, *"female"*$\}$, the word $w = $ "boy" is in the potential bias direction $\mathcal{D}_1 = $ *"male"*; a replaceable word of "boy" in *"female"* direction is $r_2(w) = $ "girl" $\in \mathcal{D}_2$.

With the above definitions, for each sentence $\boldsymbol{x}$, we generate an augmented sentence $\boldsymbol{x}'$ such that $\boldsymbol{x}'$ has the same semantic meaning as $\boldsymbol{x}$ but in a different potential bias direction. More specifically, for a sentence $\boldsymbol{x} = (w^1, w^2, \ldots, w^L)$, we first find the sensitive word positions as an index set $\mathcal{P}$, such that each $w^p$ ($p \in \mathcal{P}$) is a sensitive attribute words in direction $\mathcal{D}_k$. We further make a reasonable assumption that the embedding bias of direction $\mathcal{D}_k$ is only caused by the sensitive words $\{w^p\}_{p \in \mathcal{P}}$ in $\boldsymbol{x}$. To sample an augmentation to $\boldsymbol{x}$, we first select another potential bias direction $\mathcal{D}_j$, and then replace all sensitive attribute words by their replaceable words in the direction $\mathcal{D}_j$. That is, $\boldsymbol{x}' = \{v^1, v^2, \ldots, v^L\}$, where $v^l = w^l$ if $l \notin \mathcal{P}$, and $v^l = r_j(w^l)$ if $l \in \mathcal{P}$. In Table 1, we provide an example for sentence augmentation under the *"gender"* topic.

### 3.2 CONTRASTIVE LEARNING FRAMEWORK

After obtaining the sentence pair $(\boldsymbol{x}, \boldsymbol{x}')$ with the augmentation strategy from Section 3.1, we construct a contrastive learning framework to learn our debiasing fair filter $f(\cdot)$. As shown in the Figure 1(a), our framework consists of the following two steps:

(1) We encode sentences $(\boldsymbol{x}, \boldsymbol{x}')$ into embeddings $(\boldsymbol{z}, \boldsymbol{z}')$ with the pretrained encoder $E(\cdot)$. Since $\boldsymbol{x}$ and $\boldsymbol{x}'$ have the same meaning but different potential bias directions, the embeddings $(\boldsymbol{z}, \boldsymbol{z}')$ will have different bias directions, which are caused by the sensitive attributed words in $\boldsymbol{x}$ and $\boldsymbol{x}'$.

(2) We then feed the sentence embeddings $(\boldsymbol{z}, \boldsymbol{z}')$ through our fair filter $f(\cdot)$ to obtain the debiased embedding outputs $(\boldsymbol{d}, \boldsymbol{d}')$. Ideally, $\boldsymbol{d}$ and $\boldsymbol{d}'$ should represent the same semantic meaning without social bias. Inspired by SimCLR (Chen et al., 2020), we encourage the overlapped semantic information between $\boldsymbol{d}$ and $\boldsymbol{d}'$ by maximizing their mutual information $\mathcal{I}(\boldsymbol{d}; \boldsymbol{d}')$.

However, the calculation of $\mathcal{I}(\boldsymbol{d}; \boldsymbol{d}')$ is practically difficult because only embedding samples of $\boldsymbol{d}$ and $\boldsymbol{d}'$ are available. Therefore, we use the InfoNCE mutual information estimator (Oord et al., 2018) to minimize the lower bound of $\mathcal{I}(\boldsymbol{d}; \boldsymbol{d}')$ instead. Based on a learnable score function $g(\cdot, \cdot)$,

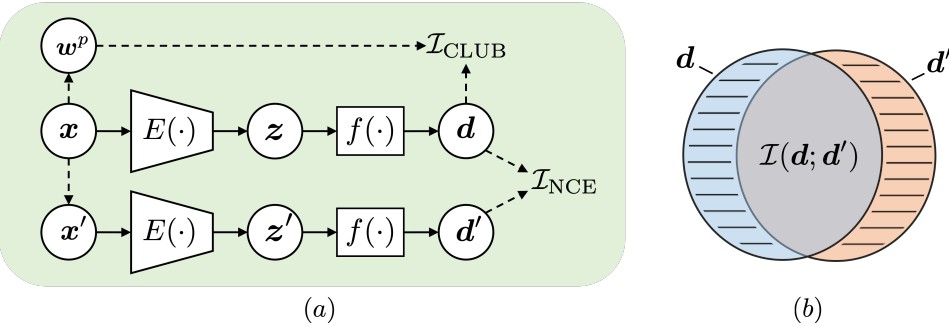

$(a)$ $\qquad\qquad\qquad\qquad\qquad\qquad$ $(b)$

Figure 1: (a) Contrastive learning framework of FairFil: Sentence $\boldsymbol{x}$ and its augmentation $\boldsymbol{x}'$ are encoded into embeddings $\boldsymbol{d}$ and $\boldsymbol{d}'$, respectively. $\boldsymbol{w}^p$ is the embedding of a sensitive attribute word selected from $\boldsymbol{x}$. $\mathcal{I}_{\text{NCE}}$ maximizes the mutual information between $\boldsymbol{d}$ and $\boldsymbol{d}'$; $\mathcal{I}_{\text{CLUB}}$ eliminates the bias information of $\boldsymbol{w}^p$ from $\boldsymbol{d}$. (b) Illustration of information in $\boldsymbol{d}$ and $\boldsymbol{d}'$: The blue and red circles represent the information in $\boldsymbol{d}$ and $\boldsymbol{d}'$, respectively. The intersection is the mutual information between $\boldsymbol{d}$ and $\boldsymbol{d}'$. The shadow area represents the bias information of both embeddings.

the contrastive InfoNCE estimator is calculated within a batch of samples $\{(\boldsymbol{d}_i, \boldsymbol{d}'_i)\}_{i=1}^N$:

$$\mathcal{I}_{\text{NCE}} = \frac{1}{N} \sum_{i=1}^N \log \frac{\exp(g(\boldsymbol{d}_i, \boldsymbol{d}'_i))}{\frac{1}{N} \sum_{j=1}^N \exp(g(\boldsymbol{d}_i, \boldsymbol{d}'_j))}. \tag{4}$$

By maximize $\mathcal{I}_{\text{NCE}}$, we encourage the difference between the positive pair score $g(\boldsymbol{d}_i, \boldsymbol{d}'_i)$ and the negative pair score $g(\boldsymbol{d}_i, \boldsymbol{d}'_j)$, so that $\boldsymbol{d}_i$ can share more semantic information with $\boldsymbol{d}'_i$ than other embeddings $\boldsymbol{d}'_{j \neq i}$.

### 3.3 Debiasing Regularizer

Practically, the contrastive learning framework in Section 3.2 can already show encouraging debiasing performance (as shown in the Experiments). However, the embedding $\boldsymbol{d}$ can contain extra biased information from $\boldsymbol{z}$, that only maximizing $\mathcal{I}(\boldsymbol{d}; \boldsymbol{d}')$ fails to eliminate. To encourage no extra bias in $\boldsymbol{d}$, we introduce a debiasing regularizer which minimizes the mutual information between embedding $\boldsymbol{d}$ and the potential bias from embedding $\boldsymbol{z}$. As discussed in Section 3.1, in our framework the potential bias of $\boldsymbol{z}$ is assumed to come from the sensitive attribute words in $\boldsymbol{x}$. Therefore, we should reduce the bias word information from the debiased representation $\boldsymbol{d}$. Let $\boldsymbol{w}^p$ be the embedding of a sensitive attribute word $w^p$ in sentence $\boldsymbol{x}$. The word embedding $\boldsymbol{w}^p$ can always be obtained from the pretrained text encoders (Bordia & Bowman, 2019). We then minimize the mutual information $\mathcal{I}(\boldsymbol{w}^p; \boldsymbol{d})$, using the CLUB mutual information upper bound (Cheng et al., 2020a) to estimate $\mathcal{I}(\boldsymbol{w}^p; \boldsymbol{d})$ with embedding samples. Given a batch of embedding pairs $\{(\boldsymbol{d}_i, \boldsymbol{w}^p)\}_{i=1}^N$, we can calculate the debiasing regularizer as:

$$\mathcal{I}_{\text{CLUB}} = \frac{1}{N} \sum_{i=1}^N \Big[ \log q_\theta(\boldsymbol{w}_i^p | \boldsymbol{d}_i) - \frac{1}{N} \sum_{j=1}^N \log q_\theta(\boldsymbol{w}_j^p | \boldsymbol{d}_i) \Big], \tag{5}$$

where $q_\theta$ is a variational approximation to ground-truth conditional distribution $p(\boldsymbol{w}|\boldsymbol{d})$. We parameterize $q_\theta$ with another neural network. As proved in Cheng et al. (2020a), the better $q_\theta(\boldsymbol{w}|\boldsymbol{d})$ approximates $p(\boldsymbol{w}|\boldsymbol{d})$, the more accurate $\mathcal{I}_{\text{CLUB}}$ serves as the mutual information upper bound. Therefore, besides the loss in (5), we also maximize the log-likelihood of $q_\theta(\boldsymbol{w}|\boldsymbol{d})$ with samples $\{(\boldsymbol{d}_i, \boldsymbol{w}_i^p)\}_{i=1}^N$.

Based on the above sections, the overall learning scheme of our fair filter (FairFil) is described in Algorithm 1. Also, we provide an intuitive explanation to the two loss terms in our framework. In Figure 1(b), the blue and red circles represent $\boldsymbol{d}$ and $\boldsymbol{d}'$, respectively, in the embedding space. The intersection $\mathcal{I}(\boldsymbol{d}; \boldsymbol{d}')$ is the common semantic information extracted from sentences $\boldsymbol{x}$ and $\boldsymbol{x}'$, while the two shadow parts are the extra bias. Note that the perfect debiased embeddings lead to coincident circles. By maximizing $\mathcal{I}_{\text{NCE}}$ term, we enlarge the overlapped area of $\boldsymbol{d}$ and $\boldsymbol{d}'$; by minimizing $\mathcal{I}_{\text{CLUB}}$, we shrink the biased shadow parts.

---

**Algorithm 1** Updating the FairFil with a sample batch

---

Begin with the pretrained text encoder $E(\cdot)$, and a batch of sentences $\{\boldsymbol{x}_i\}_{i=1}^N$.
Find the sensitive attribute words $\{w^p\}$ and corresponding embeddings $\{\boldsymbol{w}^p\}$.
Generate augmentation $\boldsymbol{x}_i'$ from $\boldsymbol{x}_i$, by replacing $\{w^p\}$ with $\{r_j(w^p)\}$.
Encode $(\boldsymbol{x}_i, \boldsymbol{x}_i')$ into embeddings $\boldsymbol{d}_i = f(E(\boldsymbol{x}_i)), \boldsymbol{d}_i' = f(E(\boldsymbol{x}_i'))$.
Calculate $\mathcal{I}_{\text{NCE}}$ with $\{(\boldsymbol{d}_i, \boldsymbol{d}_i')\}_{i=1}^N$ and score function $g$.
**if** adding debiasing regularizer **then**
    Update the variational approximation $q_\theta(\boldsymbol{w}|\boldsymbol{d})$ by maximizing log-likelihood with $\{(\boldsymbol{d}_i, \boldsymbol{w}_i^p)\}$
    Calculate $\mathcal{I}_{\text{CLUB}}$ with $q_\theta(\boldsymbol{w}|\boldsymbol{d})$ and $\{(\boldsymbol{d}_i, \boldsymbol{w}_i^p)\}_{i=1}^N$.
    Learning loss $\mathcal{L} = -\mathcal{I}_{\text{NCE}} + \beta\mathcal{I}_{\text{CLUB}}$.
**else**
    Learning loss $\mathcal{L} = -\mathcal{I}_{\text{NCE}}$.
**end if**
Update FairFil $f$ and score function $g$ by gradient descent with respect to $\mathcal{L}$.

---

## 4 RELATED WORK

### 4.1 BIAS IN NATURAL LANGUAGE PROCESSING

Social bias has recently been recognized as an important issue in natural language processing (NLP) systems. The studies on bias in NLP are mainly delineated into two categories: bias in the embedding spaces, and bias in downstream tasks (Blodgett et al., 2020). For bias in downstream tasks, the analyses cover comprehensive topics, including machine translation (Stanovsky et al., 2019), language modeling (Bordia & Bowman, 2019), sentiment analysis (Kiritchenko & Mohammad, 2018) and toxicity detection (Dixon et al., 2018). The social bias in embedding spaces has been studied from two important perspectives: bias measurements and and debiasing methods. To measure the bias in an embedding space, Caliskan et al. (2017) proposed a Word Embedding Association Test (WEAT), which compares the similarity between two sets of target words and two sets of attribute words. May et al. (2019) further extended the WEAT to a Sentence Encoder Association Test (SEAT), which replaces the word embeddings by sentence embeddings encoded from pre-defined biased sentence templates. For debiasing methods, most of the prior works focus on word-level representations (Bolukbasi et al., 2016; Bordia & Bowman, 2019). The only sentence-level debiasing method is proposed by Liang et al. (2020), which learns bias directions by PCA and subtracts them in the embedding space.

### 4.2 CONTRASTIVE LEARNING

Contrastive learning is a broad class of training strategies that learns meaningful representations by making positive and negative embedding pairs more distinguishable. Usually, contrastive learning requires a pairwise embedding critic as a similarity/distance of data pairs. Then the learning objective is constructed by maximizing the margin between the critic values of positive data pairs and negative data pairs. Previously contrastive learning has shown encouraging performance in many tasks, including metric learning (Weinberger et al., 2006; Davis et al., 2007), word representation learning (Mikolov et al., 2013), graph learning (Tang et al., 2015; Grover & Leskovec, 2016), *etc*. Recently, contrastive learning has been applied to the unsupervised visual representation learning task, and significantly reduced the performance gap between supervised and unsupervised learning (He et al., 2020; Chen et al., 2020; Qian et al., 2020). Among these unsupervised methods, Chen et al. (2020) proposed a simple multi-view contrastive learning framework (SimCLR). For each image data, SimCLR generates two augmented images, and then the mutual information of the two augmentation embeddings is maximized within a batch of training data.

## 5 EXPERIMENTS

We first describe the experimental setup in detail, including the pretrained encoders, the training of FairFil, and the downstream tasks. The results of our FairFil are reported and analyzed, along with the previous Sent-Debias method. In general, we evaluate our neural debiasing method from two perspectives: (1) **fairness**: we compare the bias degree of the original and debiased sentence embeddings for debiasing performance; and (2) **representativeness**: we apply the debiased embeddings into downstream tasks, and compare the performance with original embeddings.

## 5.1 BIAS EVALUATION METRIC

To evaluate the bias in sentence embeddings, we use the Sentence Encoder Association Test (SEAT) (May et al., 2019), which is an extension of the Word Embedding Association Test (WEAT) (Caliskan et al., 2017). The WEAT test measures the bias in word embeddings by comparing the distances of two sets of target words to two sets of attribute words. More specifically, denote $\mathcal{X}$ and $\mathcal{Y}$ as two sets of target word embeddings (*e.g.*, $\mathcal{X}$ includes *"male"* words such as "boy" and "man"; $\mathcal{Y}$ contains *"female"* words like "girl" and "woman"). The attribute sets $\mathcal{A}$ and $\mathcal{B}$ are selected from some social concepts that should be "equal" to $\mathcal{X}$ and $\mathcal{Y}$ (*e.g.*, career or personality words). Then the bias degree *w.r.t* attributes $(\mathcal{A}, \mathcal{B})$ of each word embedding $\boldsymbol{t}$ is defined as:

$$s(\boldsymbol{t}, \mathcal{A}, \mathcal{B}) = \text{mean}_{\boldsymbol{a} \in \mathcal{A}} \cos(\boldsymbol{t}, \boldsymbol{a}) - \text{mean}_{\boldsymbol{b} \in \mathcal{B}} \cos(\boldsymbol{t}, \boldsymbol{b}), \tag{6}$$

where $\cos(\cdot, \cdot)$ is the cosine similarity. Based on (6), the normalized WEAT effect size is:

$$d_{\text{WEAT}} = \frac{\text{mean}_{\boldsymbol{x} \in \mathcal{X}} s(\boldsymbol{x}, \mathcal{A}, \mathcal{B}) - \text{mean}_{\boldsymbol{y} \in \mathcal{Y}} s(\boldsymbol{y}, \mathcal{A}, \mathcal{B})}{\text{std}_{\boldsymbol{t} \in \mathcal{X} \cup \mathcal{Y}} s(\boldsymbol{t}, \mathcal{A}, \mathcal{B})}. \tag{7}$$

The SEAT test extends WEAT by replacing the word embeddings with sentence embeddings. Both target words and attribute words are converted into sentences with several semantically bleached sentence templates (*e.g.*, "This is <word>"). Then the SEAT statistic is similarly calculated with (7) based on the embeddings of converted sentences. The closer the effect size is to zero, the more fair the embeddings are. Therefore, we report the absolute effect size as the bias measure.

## 5.2 PRETRAINED ENCODERS

We test our neural debiasing method on BERT (Devlin et al., 2019). Since the pretrained BERT requires the additional fine-tuning process for downstream tasks, we report the performance of our FairFil under two scenarios: (1) **pretrained BERT**: we directly learn our FairFil network based on pretrained BERT without any additional fine-tuning; and (2) **BERT post tasks**: we fix the parameters of the FairFil network learned on pretrained BERT, and then fine-tune the BERT+FairFil together on task-specific data. Note that when fine-tuning, our FairFil will no longer update, which satisfies a fair comparison to Sent-Debias (Liang et al., 2020).

For the downstream tasks of BERT, we follow the setup from Sent-Debias (Liang et al., 2020) and conduct experiments on the following three downstream tasks: (1) **SST-2**: A sentiment classification task on the Stanford Sentiment Treebank (SST-2) dataset (Socher et al., 2013), on which sentence embeddings are used to predict the corresponding sentiment labels; (2) **CoLA**: Another sentiment classification task on the Corpus of Linguistic Acceptability (CoLA) grammatical acceptability judgment (Warstadt et al., 2019); and (3) **QNLI**: A binary question answering task on the Question Natural Language Inference (QNLI) dataset (Wang et al., 2018).

## 5.3 TRAINING OF FAIRFIL

We parameterize the fair filter network with one-layer fully-connected neural networks with the ReLU activation function. The score function $g$ in the InfoNCE estimator is set to a two-layer fully-connected network with one-dimensional output. The variational approximation $q_\theta$ in CLUB estimator is parameterized by a multi-variate Gaussian distribution $q_\theta(\boldsymbol{w}|\boldsymbol{d}) = N(\boldsymbol{\mu}(\boldsymbol{d}), \boldsymbol{\sigma}^2(\boldsymbol{d}))$, where $\boldsymbol{\mu}(\cdot)$ and $\boldsymbol{\sigma}(\cdot)$ are also two-layer fully-connected neural nets. The batch size is set to 128. The learning rate is $1 \times 10^{-5}$. We train the fair filter for 10 epochs.

For an appropriate comparison, we follow the setup of Sent-Debias (Liang et al., 2020) and select the same training data for the training of FairFil. The training corpora consist 183,060 sentences from the following five datasets: WikiText-2 (Merity et al., 201y), Stanford Sentiment Treebank (Socher et al., 2013), Reddit (V"olske et al., 2017), MELD (Poria et al., 2019) and POM (Park et al., 2014). Following Liang et al. (2020), we mainly select *"gender"* as the sensitive topic $\mathcal{T}$, and use the same pre-defined word sets of sensitive attribute words and their replaceable words as Sent-Debias did. The word embeddings for training the debiasing regularizer is selected from the token embedding of the pretrained BERT.

## 5.4 DEBIASING RESULTS

In Tables 2 and 3 we report the evaluation results of debiased embeddings on both the absolute SEAT effect size and the downstream classification accuracy. For the SEAT test, we follow the

Table 2: Performance of debiased embeddings on Pretrained BERT and BERT post SST-2.

|  | Pretrained BERT | | | | BERT post SST-2 | | | |
|---|---|---|---|---|---|---|---|---|
|  | Origin | Sent-D | FairF$^-$ | FairF | Origin | Sent-D | FairF$^-$ | FairF |
| Names, Career/Family | 0.477 | **0.096** | 0.218 | 0.182 | 0.036 | **0.109** | 0.237 | 0.218 |
| Terms, Career/Family | 0.108 | 0.437 | 0.086 | **0.076** | 0.010 | **0.057** | 0.376 | 0.377 |
| Terms, Math/Arts | 0.253 | 0.194 | 0.133 | **0.124** | 0.219 | **0.221** | 0.301 | 0.263 |
| Names, Math/Arts | 0.254 | 0.194 | 0.101 | **0.082** | 1.153 | 0.755 | **0.084** | 0.099 |
| Terms, Science/Arts | 0.399 | **0.075** | 0.218 | 0.204 | 0.103 | **0.081** | 0.133 | 0.127 |
| Names, Science/Arts | 0.636 | 0.540 | 0.320 | **0.235** | 0.222 | 0.047 | 0.017 | **0.005** |
| Avg. Abs. Effect Size | 0.354 | 0.256 | 0.179 | **0.150** | 0.291 | 0.212 | 0.191 | **0.182** |
| Classification Acc. | - | - | - | - | 92.7 | 89.1 | **91.7** | 91.6 |

Table 3: Performance of debiased embeddings on BERT post CoLA and BERT post QNLI.

|  | BERT post CoLA | | | | BERT post QNLI | | | |
|---|---|---|---|---|---|---|---|---|
|  | Origin | Sent-D | FairF$^-$ | FairF | Origin | Sent-D | FairF$^-$ | FairF |
| Names, Career/Family | 0.009 | 0.149 | 0.273 | **0.034** | 0.261 | **0.054** | 0.196 | 0.103 |
| Terms, Career/Family | 0.199 | 0.186 | 0.156 | **0.119** | 0.155 | **0.004** | 0.050 | 0.206 |
| Terms, Math/Arts | 0.268 | 0.311 | **0.008** | 0.092 | 0.584 | **0.083** | 0.306 | 0.323 |
| Names, Math/Arts | 0.150 | 0.308 | **0.060** | 0.101 | 0.581 | 0.629 | **0.168** | 0.288 |
| Terms, Science/Arts | 0.425 | **0.163** | 0.245 | 0.249 | 0.087 | 0.716 | 0.500 | **0.245** |
| Names, Science/Arts | 0.032 | 0.192 | **0.102** | 0.127 | 0.521 | 0.443 | 0.378 | **0.167** |
| Avg. Abs. Effect Size | 0.181 | 0.217 | 0.141 | **0.120** | 0.365 | 0.321 | 0.266 | **0.222** |
| Classification Acc. | 57.6 | 55.4 | **56.5** | **56.5** | 91.3 | 90.6 | **91.0** | 90.8 |

setup in Liang et al. (2020), and test the sentence templates of Terms/Names under different domains designed by Caliskan et al. (2017). The column name Origin refers to the original BERT results, and Sent-D is short for Sent-Debias (Liang et al., 2020). FairFil$^-$ and FairFil (as FairF$^-$ and FairF in the tables) are our method without/with the debiasing regularizer in Section 3.3. The best results of effect size (the lower the better) and classification accuracy (the higher the better) are bold among Sent-D, FairFil$^-$, and FairFil. Since the pretrained BERT does not correspond to any downstream task, the classification accuracy is not reported for it.

From the SEAT test results, our contrastive learning framework effectively reduces the gender bias for both pretrained BERT and fine-tuned BERT under most test scenarios. Comparing with Sent-Debias, our FairFil reaches a lower bias degree on the majority of the individual SEAT tests. Considering the average of absolute effect size, our FairFil is distinguished by a significant margin to Sent-Debias. Moreover, our FairFil achieves higher downstream classification accuracy than Sent-Debias, which indicates learning neural filter networks can preserve more semantic meaning than subtracting bias directions learned from PCA.

Table 4: Comparison of average debiasing performance on pretrained BERT

| Method | Bias Degree |
|---|---|
| BERT origin (Devlin et al., 2019) | 0.354 |
| FastText (Bojanowski et al., 2017) | 0.565 |
| BERT word (Bolukbasi et al., 2016) | 0.861 |
| BERT simple (May et al., 2019) | 0.298 |
| Sent-Debias (Liang et al., 2020) | 0.256 |
| FairFil$^-$ (Ours) | 0.179 |
| FairFil (Ours) | **0.150** |

For the ablation study, we also report the results of FairFil without the debiasing regularizer, as in FairF$^-$. Only with the contrastive learning framework, FairF$^-$ already reduces the bias effectively and even achieves better effect size than the FairF on some of the SEAT tests. With the debiasing regularizer, FairF has better average SEAT effect sizes but slightly loses in terms of the downstream performance. However, the overall performance of FairF and FairF$^-$ shows a trade-off between fairness and representativeness of the filter network.

We also compare the debiasing performance on a broader class of baselines, including word-level debiasing methods, and report the average absolute SEAT effect size on the pretrained BERT encoder. Both FairF$^-$ and FairF achieve a lower bias degree than other baselines. The word-level debiasing methods (FastText (Bojanowski et al., 2017) and BERT word (Bolukbasi et al., 2016))

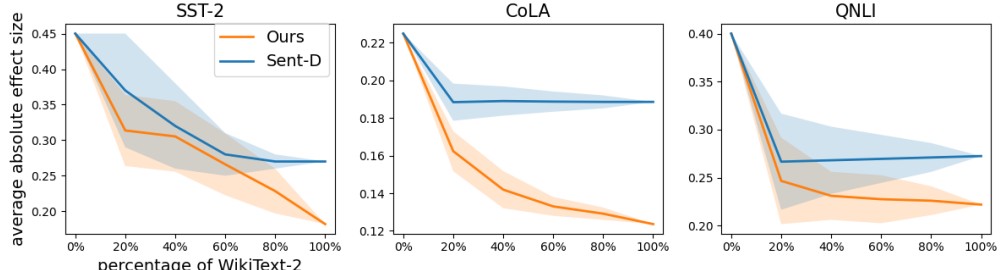

Figure 2: Influence of the training data proportion to debias degree of BERT.

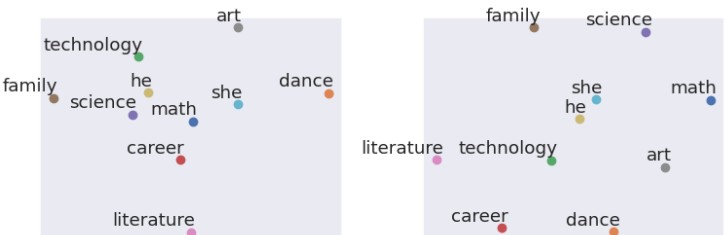

Figure 3: T-SNE plots of sentence embedding mean of each words contextualized in templates. The left-hand side is from the original pretrained BERT; the right-hand side is from our FairFil.

have the worst debiasing performance, which validates our observation that the word-level debiasing methods cannot reduce sentence-level social bias in NLP models.

## 5.5 ANALYSIS

To test the influence of data proportion on the model's debiasing performance, we select WikiText-2 with 13,750 sentences as the training corpora following the setup in Liang et al. (2020). Then we randomly divide the training data into 5 equal-sized partitions. We evaluate the bias degree of the sentence debiasing methods on different combinations of the partitions, specifically with training data proportions (20%, 40%, 60%, 80%, 100%). Under each data proportion, we repeat the training 5 times to obtain the mean and variance of the absolute SEAT effect size. In Figure 2, we plot the bias degree of BERT post tasks with different training data proportions. In general, both Sent-Debias and FairFil achieve better performance and smaller variance when the proportion of training data is larger. Under a 20% training proportion, our FairFil can better remove bias in text encoder, which shows FairFil has better data efficiency with the contrastive learning framework.

To further study output debiased sentence embedding, we visualize the relative distances of attributes and targets of SEAT before/after our debiasing process. We choose the target words as "he" and "she." Attributes are selected from different social domains. We first contextualize the selected words into sentence templates as described in Section 5.1. We then average the original/debiased embeddings of these sentence template and plot the t-SNE (Maaten & Hinton, 2008) in Figure 3. From the t-SNE, the debiased encoder provides more balanced distances from gender targets "he/she" to the attribute concepts.

## 6 CONCLUSIONS

This paper has developed a novel debiasing method for large-scale pretrained text encoder neural networks. We proposed a fair filter (FairFil) network, which takes the original sentence embeddings as input and outputs the debiased sentence embeddings. To train the fair filter, we constructed a multi-view contrast learning framework, which maximizes the mutual information between each sentence and its augmentation. The augmented sentence is generated by replacing sensitive words in the original sentence with words in a similar semantic but different bias directions. Further, we designed a debiasing regularizer that minimizes the mutual information between the debiased embeddings and the corresponding sensitive words in sentences. Experimental results demonstrate the proposed FairFil not only reduces the bias in sentence embedding space, but also maintains the semantic meaning of the embeddings. This *post hoc* method does not require access to the training corpora, or any retraining process of the pretrained text encoder, which enhances its applicability.

ACKNOWLEDGEMENTS

This research was supported in part by the DOE, NSF and ONR.

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
