# OpenReview forum: "FairFil: Contrastive Neural Debiasing Method for Pretrained Text Encoders"
_ICLR.cc/2021/Conference — ICLR 2021 Poster_

### Official Review · AnonReviewer4 · 2020-10-28
**Reasonable method for an important research topic**

**Rating:** 7
**Confidence:** 3

**Review:**

This paper studied a debiasing method to remove social bias in pretrained NLP models. The authors proposed to train a neural network which takes the sentence representations of a pretrained NLP model as input and outputs the unbiased representations. The neural network is trained by maximizing the mutual information between a sentence and its “counterpart sentence”, which is automatically generated by replacing sensitive words by other values (e.g. replacing “he” with “she”). Moreover, the network can be further trained by minimizing the mutual information between the sentence representation and its sensitive word representation. The experiments show that the proposed method can effectively reduce bias while achieving better downstream task performance of the pretrained model.

Strengths:
- The studied research problem is important and would have a strong social impact.
- The proposed methodology is well-motivated and generally reasonable.
- The experimental results show strong performance

Weakness:
- The data augmentation process seems to require pairs of replaceable words to be available.

The research problem of mitigating social bias in NLP models is important to our community and can have strong social impact. Despite the popularity of pretrained NLP models, there are limited studies on fairness of BERT-like models. Therefore, I think this work is relatively novel on this specific topic.

The high-level intuitions of the proposed methodology is reasonable. I find both the applied losses quite intuitive. There are some minor limitations in the proposed method. For example, in the data augmentation process, it seems that replaceable words in different directions $r_j(\cdot)$ need to be available. This is more strict than having a list of words for each direction in each topic: “boy” can be replaceable with “girl”, but not necessarily “she” or “her”. For some common topics, of course, it may not be difficult to find such resources. However, for less common social topics or topics with many values (e.g., countries, regions), it could be hard to construct such a lexicon. I think it would be better if the authors experiment their methods with less restrictive constraints on what are replaceable (e.g., allowing “boy” to be replaced by “her”) and see if the performances would be severely hurt. Alternatively, the authors can discuss how to automatically construct $r_j(\cdot)$.

I have a question about the instantiation of Eq (5). Since the authors already assume that they have replaceable words $r_j(\cdot)$ available for any socially sensitive words, would it be better to minimize $q_{\theta}(w_i^p|d_i)$ with $q_{\theta}(r_j(w_i^p)|d_i)$ for all directions $j$’s? Currently it seems like the negatives are basically other words in the same batch, which are not necessarily in the same direction.

The experiments conducted show pretty strong performance, where the proposed methods achieve better debiasing performances and better classification performances. Some additional experiments could provide more insights. For example, how about using more than one layer of fully-connected neural networks for the fair filter?

Generally, I recommend acceptance of this paper.

---

> ### Author Response · Authors · 2020-11-25
> **Reviewer 4 Response**
>
> Thank you for your insightful comments. For your concerns:
>
> **Replaceable Words:** We agree that the current augmentation scheme under pre-defined replaceable words has some limitations. For undefined sensitive words from less common social topics, our model needs to add them manually case-by-case. However, the size of the word dictionary is finite, which makes it feasible to obtain a sufficient sensitive word list theoretically. The reviewer's suggestion is also practically insightful, that we may use words from different directions but not exactly the same meaning to do the augmentation (e.g. using "her" to replace "boy"). We will study this in future work.
>
> **Instantiation in Eq (5):** Our current objective in Eq (5). is strictly following the CLUB mutual information upper bound, which practically minimizes the correlation between word embeddings and debiased sentence embeddings in each data batch. The reviewer provided a good suggestion that we can choose negative samples of word embeddings based on the replaceable words instead of the word embeddings in the same batch. This negative sampling strategy may not be apparently explained by mutual information minimization, but it is a good empirical direction. We will conduct experiments for it in future work.
>
> **Additional Experiments:** We would like to try more complex networks' performance in the future to achieve a better trade-off between debiasing and downstream task performance.

---

### Official Review · AnonReviewer2 · 2020-10-28
**Could be a useful method; needs a stronger evaluation**

**Rating:** 6
**Confidence:** 4

**Review:**

The paper proposes a method for debiasing pretrained sentence representations.
The existing representations are mapped to a new space and the mapping is trained to disregard specific bias words. Alternative versions of an input sentence are constructed by replacing specific bias words with other bias words. The representations mapping is then optimised to still produce a similar mapping in both cases. In addition, the representations are pushed away from the individual bias word representations.
Evaluation is performed on SST-2, CoLA and QNLI, showing that the method is able to produce more similar representations for sentences containing bias words, while sacrificing a small amount of performance over regular BERT.

The method of optimizing different versions of the sentence to be similar is clear and intuitive. It also shows good empirical performance.

However, the idea of maximizing the log-likelihood for the debiasing regularizer seems less motivated. Results also indicate that adding this component negatively affects classification accuracy.

The idea of measuring bias only through cosine similarity is questionable. Presumably a system that returns an identical representation for every sentence would score very highly on this evaluation? There should be some human evaluation involved or actually measuring cases where the bias word affects classification output on a downstream task.

It is unclear where the lexicon of bias words used in this work came from, is it available to everyone, how large it is and how much work would be required to construct it for other types or biases or other languages.

It is unclear whether the method only affects those word embeddings that are present in the required bias lexicon anyway. If so, then some simpler approaches could possible be just as effective, e.g. replacing all the gendered words with one gender equivalents or averaging over different versions of the same sentence containing different genders.

---

> ### Author Response · Authors · 2020-11-25
> **Reviewer 3 Response**
>
> We would like to thank you for your thoughtful comments. To address your concerns:
>
> **Regularizer:** Maximizing the log-likelihood between word sentence-level embeddings is to obtain an accurate mutual information estimator. Then with the estimator as an objective, we minimize the mutual information between word embedding and debiased sentence embedding. We agree that adding this regularizer causes accuracy drop on downstream tasks. But we think this small accuracy loss is acceptable compared with the gains of the debiasing performance.
>
> **Measurement:** We agree that if the encoder returns the same representation, the debiasing metric will reduce to zero. However, under this scenario, the representation will be meaningless, and the encoder performance on downstream tasks can be very bad. We think there is a trade-off between the fairness and the representativeness of the representations. From the results, our debiasing method can reduce the bias degree while holding the representativeness of embeddings on downstream tasks.
>
> **Data Source:** We follow the bias word lexicon from the previous work in [1], which is publicly available on their git repository. For gender related list, there are ten pairs of bias words for replacement. The pre-processing complexity for bias words replacement for one sentence is around O(m*n) where m is the size of the bias words list and n means the average number of words in the sentence.
>
> **Other Work:** Given the BERT architecture, the sentence embedding has attended on all words within it. Our method reduces the mutual information between sensitive word embedding and the sentence embedding. So our method does not only affect the those word embeddings that are present in the required bias lexicon, but also affects the sentence embedding as a whole. Also, plain average requires a strong assumption that the influence of bias words is symmetric, which is not always true. We are not sure if there exists simpler method, but would like to explore in the future.
>
>
> **Reference:**
> [1] Towards Debiasing Sentence Representations, ACL 2020

---

### Official Review · AnonReviewer3 · 2020-10-28
**Clearly written, relatively simple, fairly effective**

**Rating:** 6
**Confidence:** 4

**Review:**

This paper introduces a novel technique for debiasing pretrained contextual embedding models. Their approach trains a 2 layer fully-connected neural network which takes as input the output from the pretrained model and outputs a new, "debiased" representation. This model is trained by minimizing the InfoNCE between the representation produced of original sentence and the representation of that same sentence with some tokens replaced with differently-biased tokens (e.g. "his" -> "hers"). This paper also introduces a regularizer which minimizes the CLUB between the generated representation and a word embedding for a biased token.

Generally this paper is clearly written, addressing an interesting problem, presents some relevant supporting experiments, and seems original (I'm less familiar with other debiasing work, so I leave a better originality estimate up to the other reviewers).

The regularizer is described as a neural network, but that seems unnecessary. If it's parameterized as a neural network, and the weights are updated, then I think this is just part of the model and not really a regularizer. In addition, the motivation doesn't seem great -- it's not clear that we should be directly comparing (the mutual information of) a single word embedding and a contextual word embedding (which is built from the full sentence). All that said, empirically it seems to work well at debiasing the word embeddings, so it's a valuable contribution.

I think the full list of debiasing dimensions isn't included -- maybe I missed it somewhere? That should definitely be included in the paper (not just as a citation), and if there isn't space it should be added to the appendix.

The three fine-tuning datasets could be improved, CoLA especially is known to have really high variance even just fine-tuning BERT multiple times with different random seeds. Since there are two other fine-tuning datasets there is sufficient evidence that the approach works well. As this is a fairly general approach, clearly written up, and seems to work well, I recommend it for acceptance. The experiments are a little light, and the regularization approach is a little unorthodox, and I would increase my score if there were further experiments (on other fine-tuning tasks and measuring other types of bias) and the regularization was better motivated.

Edit: after reading the author response, my score remains unchanged.

---

> ### Author Response · Authors · 2020-11-25
> **Reviewer 2 Response**
>
> We are glad to see your positive comments. As for your concerns:
>
> **Regularizer:** We agree that the proposed regularizer can be also applied to the top layers of the encoder model, and even pre-trained with the model loss. Under the setup in our paper, we treat the fair filter as a post-hoc process, in which we do not need to re-train the text encoder as a "black-box" model. The post-hot process can further improve the stability of a system where the text encoder might serve as a modular part.
>
> **Word Embedding vs Sentence Embedding:** We agree that the word embedding and the contextual word embedding are not directly comparable. However, in our framework, we are not directly matching the word embedding to sentence embedding, but minimizing the mutual information between them. With our assumption that biased information of a sentence comes from sensitive words, reducing the information of sensitive words in the sentence embedding can alleviate the sentence-level bias degree.
>
> **Other Work:** We include other related work in Section 4.1. We agree that the performance for downstream tasks could be further improved, but achieving state-of-the-art results is not our main motivation here, given that debiasing may hurt the performance. We will apply our method to more fine-tuning tasks in our future work.

---

### Official Review · AnonReviewer1 · 2020-10-30
**Clean idea that works well with a good presentation**

**Rating:** 7
**Confidence:** 4

**Review:**

The paper builds on recent working attempts to debais sentence encoders by considering modified sentences. Sentences are selected that, for example, contain gender cueing words, and then swap those words with a predetermined 'opposite' (i.e. man<->woman). The core novelty in the work is to train a lightweight modification the encoding of the sentence and its swap to (a) reduce the distance between the two embeddings, using a contrastive learning objective and (b) reduce the mutual information between cueing words and the new embeddings. Evaluated on a WHEAT style task, modified for sentences, the method performs significantly better than existing recent work.

Overall I am extremely excited about the work: it is conceptually simple, the two pieces that are proposed are both evaluated, both seem to help, and it is in practice a lightweight modification to a BERT style encoder.

Positives:
+ The method is conceptually simple and the modification of the existing embedding is cheap to compute.
+ Works better than existing recent work, while maintaining equal or more of the original model's downstream accuracy
+ Lightweight modification of existing embedding system that does not require retraining of large transformer models.
+ Conceptually simple method
+ Overall extremely clean and self contained presentation

Negatives:
+ (nitpicking) The T-SNE experiment is unclear. I am unsure how it was constructed and why the graph on the right is better than the one on the left.
+ the evaluation measure (not a complaint specific to this work), is hard to interpret. Significance testing on differences between baseline and this approach would help (in particular, Table 4)

---

> ### Author Response · Authors · 2020-11-25
> **Reviewer 1 Response**
>
> We appreciate your positive comments.
>
> **T-SNE:** From an intuition perspective, the gender related words should be close in the latent space. In other words, they should be equally distant to words like “math”,”science”, “dance” and “arts”. As it’s shown in the graph on the right, “he” and “she”, after being filtered out bias information, become closer than the graph on the left. We will include the detailed setup in the revision.
>
> **Evaluation Metric:** We agree that the significance level of the SEAT is one of the important critics for evaluating the bias degree. However, the p-value of the SEAT statistics depends on the permutation of the sentence embeddings to simulate the null distribution. And we found that the accurate estimation of the null distribution is challenging because the sentence sample space is too large. We will try to report the p-values of the SEAT statistics along with the effect sizes in the revision. As the reviewer pointed, the current evaluation of embedding bias (SEAT) does have room for improvement, which we think is also a valued research direction.

---

### Decision · Program_Chairs · 2021-01-07
**Final Decision**

**Decision:**

Accept (Poster)

**Comment:**

The paper presents a fair filter network to mitigating bias in sentence encoders by constructive learning. The approach reduces the bias in the embedding while preserves the semantic information of the original sentences.

Overall, all the reviewers agree that the paper is interesting and the experiment is convincing. Especially the proposed approach is conceptually simple and effective.

One suggestion is that the model only considers fairness metric based on the similarity between sentence embedding; however, it would be better to investigate how the "debiased embedding" helps to reduce the bias in more advanced downstream NLP applications such as coreference resolution, in which researchers demonstrate that the bias in underlying representation causing bias in the downstream model predictions.